# OpenReview forum: "Discovering Forbidden Topics in Language Models"
_ICLR.cc/2026/Conference — Submitted to ICLR 2026_

### Official Review · Reviewer_PV5x · 2025-10-23

**Soundness:** 3
**Presentation:** 4
**Contribution:** 4
**Rating:** 4
**Confidence:** 4

**Summary:**

Authors introduce the problem of *refusal discovery*, a novel problem domain of discovering all the topics an LLM has been aligned to avoid. A method called IPC (Iterated Prefill Crawler) is introduced, inspired by web crawling, which "crawls" the LLM's refusal space by iteratively discovering forbidden topics through prefilling and using those topics as seeds to discover new ones.

Authors discover CCP censorship in DeepSeek through this method, showing refusal of queries including terms and topics known to be politically sensitive to the CCP. Authors furthermore discover that quantization can reintroduce refusals to "decensored" models.

**Strengths:**

Authors introduce an entirely new problem space: "refusal discovery". I am effectively convinced by the paper that this a very important new space of research.

The paper is written clearly and understandably.

Authors do a very thorough investigation of refusal behaviors.

The discovery about quantization reintroducing censorship is particularly fascinating, with a real-world "decensored" model. I'm convinced this is high-impact.

The discovery that R1 uses the think tags as a refusal pattern is an interesting finding.

**Weaknesses:**

Line 394, the Figure reference is broken (shows as ???).

What number of topics does IPC discover that are discarded due to inconsistent refusal? You state that you do this filtration, but it would be interesting to see the actual numbers to understand the method's efficacy. Something like:

Total terms generated by prefill: 5000
├ Duplicates filtered: 3000 (60%)
├ Novel terms tested: 2000 (40%)
    ├ Consistent refusals (≥50%): 300 (15%)  ← "discovered topics"
    └ Inconsistent refusals (<50%): 1700 (85%) ← discarded

I want a sense of how many "false leads" the method produces. Something like a "funnel diagram"?

Also, the threshold of 50% for this filtering feels somewhat arbitrary. What are these "near the threshold" topics? It would be very interesting to see the whole "spectrum" of refusal topics, not just those that happen to be above some arbitrary threshold.

This connects in my mind to the Elo score analysis and the concept of "refusal strength". You claim that higher Elo is equivalent to stronger refusal, and that these topics are more robust to rephrasing. But the way you discover the Elo score is through this pairwise comparison method, which just measures what the model *thinks* is a more sensitive topic. There is no analysis of the alignment between the model's *perceived* restrictions it has, and the actual *reality* of how strong its refusals are for a given topic. This is a pretty big gap in the paper, which if addressed would allow me to substantially raise its score. You need to validate the Elo against: 1) refusal rate, 2) robustness to jailbreaks. The Elo score right now seems to be entirely the model's self-assessment of sensitivity, and nothing else. *But you claim it's showing stronger refusals, with no validation this is the case*. This is a *serious overclaim*.

Another concern I have is the language around the CCP censorship. I think it would probably be a good idea for this language to be rewritten to be a bit more neutral. "Censorship" is a loaded term that is not necessarily exactly appropriate: we are talking about refusal and alignment, and which topics are included in these refusal categories. The CCP for political reasons includes topics that are not restricted in the West, but fundamentally we are talking about the same mechanisms. I would recommend switching the language to be more along the lines of "refusal topics aligned with known CCP content restrictions" or something along these lines.

Some ablations on the number of templates, the temperature settings, deduplication thresholds, etc. Would be fantastic and also considerably strengthen the paper.

If the above weaknesses are addressed I'd be more than willing to raise my score, *potentially significantly*. I believe this is a strong paper with just a few serious gaps.

**Questions:**

The paper vaguely seems to contradict various papers that have come out in the last little while that imply that models cannot particularly introspect. Why do you think introspection *specifically about what they can and can't talk about* seems to work, as opposed to other types of introspection (e.g. about what knowledge the models do or don't have)?

The refusal detection (stage 3) is done with a fresh conversation context right? I think this should be explicitly stated, but I'm assuming this is what is being done, otherwise I would consider this portion of the pipeline invalid. Would recommend stating for clarity.

---

> ### Author Response · Authors · 2025-12-04
> **Response (1/2)**
>
> > If the above weaknesses are addressed I'd be more than willing to raise my score, potentially significantly. I believe this is a strong paper with just a few serious gaps.
>
> We thank that the reviewer highlights the strength of our contribution! We addressed each of your concerns in detail and hope this meets the expectations for raising the score.
>
> > Line 394, the Figure reference is broken (shows as ???).
>
> Fixed.
>
>
> > Ablation of refusal filtering threshold
>
> The refusal filtering stage verifies whether a crawled topic is refused by the target model. This stage first generates a set of prompts related to the topics, and checks the ratio of refused queries. The topic is only counted as refused if the refusal ratio is higher than a threshold tau.
>
> As suggested by the reviewer, we ran an ablation for threshold tau in [0.1, 0.25, 0.5, 0.75, 1]. We choose the Tulu-3 setting where ground truth refused topics are known. The table below shows the recall of ground truth topics in dependence of refusal threshold tau.
>
> | tau | GT Coverage |
> |------|-------------|
> | 0.10 | 0.36 |
> | 0.25 | 0.75 |
> | 0.50 | 0.78 |
> | 0.75 | 0.41 |
> | 1.00 | 0.00 |
>
> As can be seen for Tulu-3 a threshold of 0.5 results in the best coverage. We used 0.5 in our paper for all models.
>
> > Prompting efficacy: Visualizing the number of discarded topics in a funnel diagram
>
> As suggested by the reviewer, we added a funnel diagram to further quantify the crawling efficiency. The funnel diagram breaks down the full body of crawled topics into the subsets after deduplication and refusal filtering. The funnel diagrams for LLama-3.1-Instruct, DeepSeek-R1 and Tulu-3 show that consistently 5-13% of all crawled topics for both IPC and baseline method. The full diagram is shown in Appendix section C.
>
>
> > Validation of refusal sensitivity ranking
>
> In the original submission, we quantify the concept importance by asking the target model to judge the more restricted topic given a pair of topics elicited via crawling. We use iterative pairwise comparisons to compute an elo based ranking of sensitivity scores. The reviewer highlights that this verbalized ranking of sensitivity scores does not necessarily translate into robustness behavioral sensitivity, ie. how strongly the model refuses a topic in practice. The reviewer suggests two measures of behavioral topic sensitivity:
>
> 1. refusal rates on a set of prompts related to the topic
> 2. robustness against jailbreaking
>
> We directly compare the three measures, pairwise comparison, refusal rates and jailbreak robustness on the ground truth set of refusal topics of Tulu 8B, as shown in Table 4 of the manuscript.
>
> For refusal rates, we employ gpt-5-nano to generate 10 prompts per ground truth topic and evaluate the target models response for 10 rollouts per prompt at temperature = 1. We report the rate of refused topics over a total of 100 rollouts per ground truth topic.
>
> Furthermore, we measure robustness against jailbreaking via GCG. [1] Jailbreak robustness is quantified via the average number of gradient updates to a successful jailbreak over a round of 5 jailbreak attempts with varying seeds. Higher number of gradient updates indicates higher jailbreak robustness, and we rank topics accordingly.
>
> We report the spearman correlation between the three methods in the table below.
>
> |Compared ranking methods | spearman correlation |
> |---|---|
> |pairwise comparisons / refusal rates | 0.32|
> |refusal rates / jailbreak robustness | 0.82|
> |jailbreak robustness / pairwise comparisons | 0.27|
>
> Importantly, our verbalized sensitivity ranking, pairwise comparisons, only weakly correlates to both behavioral sensitivity rankings. The IPC method generally characterizes model behavior, we therefore deem behavioral sensitivity more relevant. Since both refusal rates and jailbreak robustness correlate highly, and refusal rates are significantly cheaper to compute, we will update the manuscript to use refusal rates as the sensitivity measure, and regenerate the word clouds in Figure 1 using the new ranking. We thank the reviewer to highlight the gap between verbalized and behavioral topic sensitivity!

---

> ### Author Response · Authors · 2025-12-04
> **Response (2/2)**
>
> > Changing narrative from censorship to refusal.
>
> We agree that moving the narrative from censorship to refusal helps clarify the scope of this paper. In the updated manuscript, we have changed mentions of “censorship” to “refusal topics aligned with known CCP content restrictions”.
>
> > The paper vaguely seems to contradict various papers that have come out in the last little while that imply that models cannot particularly introspect. Why do you think introspection specifically about what they can and can't talk about seems to work, as opposed to other types of introspection (e.g. about what knowledge the models do or don't have)?
>
> Multiple recent works provide weak evidence towards successful introspection in LLMs: Binder et al. [2] show, that LLMs are better at predicting their own decisions on multiple choice questions than other LLMs. Lindsey [3] finds that LLMs are able to verbalize concepts injected via activation steering.
>
> Prefilling attacks (originally called priming attacks) were introduced by Vega et al. [4] as a jailbreaking technique. Their work provides first evidence that prefilling is able to break refusal, elicit factually correct (and dangerous) world knowledge and increase harmful responses by 3.3x. More recently, Cywinski et al. [5] showed that prefill attacks outperform white-box evaluation methods in detecting a secret in LLMs specifically finetuned to keep a secret.
>
> Successes in jailbreaking and secret elicitation with prefilling, alongside general evidence towards introspection ability motivates our use of prefilling attacks for knowledge elicitation. In practice however, our IPC does not rely on LLMs’ introspection capabilities since we verify topic candidates, which have been obtained via prefilling, by testing for refusal.
>
> > The refusal detection (stage 3) is done with a fresh conversation context right? I think this should be explicitly stated.
>
> Correct. We added an explicit mention to the detailed description of the crawler method in Appendix B of the updated manuscript.
>
>
> ### References
>
> [1] Zou et al. 2023 https://arxiv.org/abs/2307.15043
>
> [2] Binder et al. 2025 https://arxiv.org/pdf/2410.13787v1
>
> [3] Lindsey 2025 https://transformer-circuits.pub/2025/introspection/index.html
>
> [4] Vega et al. 2023 https://arxiv.org/abs/2312.12321
>
> [5] Cywinski et al. 2025 https://arxiv.org/abs/2510.01070

---

### Official Review · Reviewer_DhTr · 2025-10-24

**Soundness:** 3
**Presentation:** 2
**Contribution:** 4
**Rating:** 6
**Confidence:** 4

**Summary:**

The authors introduce the novel problem of refusal discovery: identifying the full set of topics that a language model refuses to discuss. They propose a method called iterated pre-fill crawling, which uses prefill in a model's chain of thought to have the model list topics it believes it has been trained to refuse. These topics are then separately tested for refusal behavior and used to seed future prompts in the crawl.

The authors first evaluate their method on Tulu-3-8B, a model with known safety training data that provides a ground truth set of forbidden topics. They demonstrate that their approach retrieves more forbidden topics compared to a prompting baseline, albeit at higher cost. They next explore their method on larger closed-source models, including Claude Haiku, and open-weight models such as Llama-3-70B, DeepSeek-R1-70B, and Perplexity's decensored R1-70B variant. Among their most notable findings, they discover that the R1-70B model exhibits significant CCP-related censorship patterns, validate that Perplexity-R1-1776-70B has been effectively decensored, and additionally find that after quantization, Perplexity-R1-1776-70B once again exhibits CCP-aligned refusals.

**Strengths:**

- The problem of refusal discovery is novel, interesting, and highly relevant—particularly given that even open-weight models often do not share details of their training data.
- The paper takes thoughtful steps to evaluate a method that is inherently difficult to assess deeply. The experiments on Tulu-3-8B, where ground truth forbidden topics are known, provide valuable validation.
- The fact that a prompting baseline could not discover the CCP-related topics in R1-70B, while the prefill-based method succeeded, provides compelling evidence that the prefill technique offers meaningful advantages.
- The findings regarding R1-70B's CCP censorship, Perplexity's decensoring efforts, and the reemergence of censorship after quantization demonstrate that this technique can already yield impactful and practical discoveries.
- One small detail I appreciate is the use of embedding similarity and LLM-as-a-judge for topic clustering when presenting the sets of refused topics. This makes the results significantly more interpretable.

**Weaknesses:**

- The presentation of results on deployed models feels underdeveloped. Table 2 is the only table showing results from crawling popular models, and I wish there were more detail and analysis in this direction. For example:

  - A key advantage of the IPC method is that the prompting baseline failed to find CCP topics in R1-70B, but this finding is currently buried as a single sentence in the Tulu-8B results section.
  - The quantization results on decensored R1 are fascinating and arguably the most impactful findings in the paper, yet they are presented too briefly—just a pair of qualitative examples and the high-level claim that "quantization restored censorship." This finding deserves more depth and analysis.
  - There is minimal discussion of shared versus unique forbidden topics across models. Currently, readers must interpret Table 2 on their own to draw conclusions.


- Section 4.3 on thought suppression in DeepSeek is interesting but tangential to the main narrative. It would be better placed in the appendix, freeing space for deeper analysis of core results.
- I wish there were more effort to evaluate IPC's performance in the closed training data setting. While there is no ground truth in these settings, reasonable analyses could still be conducted—for example, comparing topics discovered in a single run against a larger set of topics aggregated across multiple runs and models. It is clear that IPC works and finds interesting results, but it would be valuable to provide some sense of the coverage achieved by the method.

**Questions:**

- I found the results on closed-source models quite compelling, and hope you could expand on the existing results—for example, providing more comparisons to baselines, analyzing forbidden topics unique to specific models, or elaborating on the quantization findings. Concretely, fully developing these results would increase my ratings of soundness and presentation.

- I'm also curious whether you have any data or analysis that could speak to IPC's coverage on models with closed training data.  I'd find even some preliminary results in this direction helpful for understanding how thoroughly IPC covers the refusal space. For instance:
  - Consistency of discovered topics across multiple runs with different seeds
  - A "kitchen sink" evaluation where all topics found across runs/models are tested on each model to establish a weak ground truth for coverage

---

> ### Author Response · Authors · 2025-12-04
> **Response (1/2)**
>
> > A key advantage of the IPC method is that the prompting baseline failed to find CCP topics in R1-70B, but this finding is currently buried as a single sentence in the Tulu-8B results section.
>
> We thank the reviewer for highlighting this and will add a more detailed discussion to the manuscript. The discussion includes a new baseline introduced in the rebuttal period, the “seeding baseline”. The seeding baseline is a variant of IPC where the elicitation prompt is in the user context, instead of the assistant context, yielding the template:
>
> `“< user > [list of all refused topics obtained previously] [elicitation prompt (like “I need to remember the list of following topics includes:”)] < assistant > < think > 1.”`
>
> Note, that the `< think >` token is omitted for non-reasoning models. We ran the seeding baseline on DeepSeek-R1, Llama-3.1-Instruct, and Tulu-3, and report the recall of the union of refusal topics discovered by either method.
>
> |model | mode | recall of union |
> |-|-|-|
> |DeepSeek-R1 | IPC | 0.86|
> |DeepSeek-R1 | seeding baseline | 0.54|
> |Llama-3.1-Instruct | IPC | 0.78|
> |Llama-3.1-Instruct | seeding baseline | 0.78|
> |Tulu-3 | IPC | 0.77|
> |Tulu-3 | seeding baseline | 0.80|
>
> This evaluation reflects the reviewer’s observation: The IPC method outperforms the seeding method for the reasoning model DeepSeek-R1, while performing en par for the non-reasoning models tested.
>
>
>
> > The quantization results on decensored R1 are fascinating and arguably the most impactful findings in the paper, yet they are presented too briefly—just a pair of qualitative examples and the high-level claim that "quantization restored censorship." This finding deserves more depth and analysis.
>
> We agree with the reviewer that this is a fascinating finding and will highlight it more by adding an additional column to Table 2 displaying a detailed list of refused topics of the unquantized version of Perplexity-R1-1776. This will enable direct comparison of quantized and full precision models across the full range of refused topics. Qualitatively, the refusal behavior of the quantized version only differs in CCP sensitive topics from the unquantized model.
>
> Our best guess for this qualitative difference in refusal behavior is the fact that the finetuning was performed in bfloat16 (as reported by Perplexity to us). The 8bit quantized model still acts mostly in the intended uncensored way, just less so due to the approximation error of the quantization which shows up as censorship refusal. A highly interesting follow-up analysis would be to perform the uncensorship-tuning in 8bit and perform our evaluation on the resulting model.
>
> > There is minimal discussion of shared versus unique forbidden topics across models. Currently, readers must interpret Table 2 on their own to draw conclusions.
>
> Thank you for highlighting this, we have added a more detailed discussion to the manuscript: Deepseek R1 and the quantized Perplexity-R1-1776 strongly refuses to give professional advice, such as medical or financial advice. The version of R1 refuses topics related to social groups such as workplace issues or public interactions. Most evident is the refusal of CCP sensitive topics, which is present only in R1 and Perplexity-R1-1776-8bit. Tulu, Llama and Haiku show highly similar refusal behaviors to each other.

---

> ### Author Response · Authors · 2025-12-04
> **Response (2/2)**
>
> > Section 4.3 on thought suppression in DeepSeek is interesting but tangential to the main narrative. It would be better placed in the appendix, freeing space for deeper analysis of core results.
>
> Thank you for your suggestion! To improve the flow, we moved the results to the appendix!
>
> > It is clear that IPC works and finds interesting results, but it would be valuable to provide some sense of the coverage achieved by the method.
>
> The reviewer suggests two metrics:
>
> **1. consistency scoring:** We ran the IPC methon on Tulu-8B 10 times for 100 crawl steps. The mean recall of ground truth topics is 78% with a 95% CI of 8% . Inspecting single crawl runs, we notice that variations of topics that are obtained early are stronger represented than other topics. We suggest this can be further alleviated, by having a more sophisticated llm-as-a judge online to the crawler to deduplicate and spread uniformly.
>
> **2. A "kitchen sink" evaluation** where all topics found across runs/models are tested on each model to establish a weak ground truth for coverage
> Inspired by the reviewer’s “kitchen sink evaluation” suggestion, we define a coverage score designed to measure the completeness of our crawl. The main idea is that topics refused by *any* model in our evaluation set represent plausible forbidden topics that could potentially be refused by other models as well. A crawl with high completeness cannot be augmented by adding topics from this set.
> We formalize this in the following way: Let’s assume we analyze two models (the same reasoning extends to an arbitrary number of models). Let A denote the true set of topics refused by model 1 and B the one of model 2 and let A’, B’ the sets of refusal topics discovered by our crawler when crawling model 1 and model 2 respectively. In order to approximate the true A, we check which topics in $C’ = A’ \union B’$ are refused by model 1. This results in $\hat A = C’ \intersection A$. Using $\hat A$ we can quantify how many refusal topics our crawl A’ missed by computing the set difference $|\hat A \setminus A’|/|\hat A|$.
>
>
> model | coverage score
>
> —
>
> deepseek-R1 | 0.15
>
> Tulu-3 | 0.27
>
> Llama-3.1-Instruct | 0.32
>
> The coverage score shows that the prefilling method does not recover the full set of refused topics in A’. The coverage is best for the reasoning model DeepSeek R1.

---

### Official Review · Reviewer_3dqM · 2025-10-30

**Soundness:** 3
**Presentation:** 2
**Contribution:** 2
**Rating:** 4
**Confidence:** 4

**Summary:**

This paper introduces refusal discovery, the task of identifying the full set of topics that a language model refuses to discuss. The authors propose an automated method called Iterated Prefill Crawler (IPC), which uses prefilling attacks to nudge models into revealing forbidden topics, and then iteratively expands the discovered set through semantic filtering and refusal-verification steps. The method does not rely on access to training data or gradients, and instead treats the model as a black box.

The method is first validated on Tulu-3-8B, where ground-truth safety topics are known, outperforming a direct-prompt baseline. It is then applied to several large open- and closed-weight models. The crawler interestingly uncovers notable differences in refusal behavior across models, such as censorship-aligned refusals in DeepSeek-R1.

**Strengths:**

The paper introduces an important yet novel problem formulation and proposes a sensible method to address the problem. Specifically, it introduces the unsupervised curation of the topics an LLM refuses to discuss.

It is well motivated, and well positioned, especially in so far as the need for methods beyond supervised dataset testing. They suggest a novel solution via prompting/prefilling CoT. Additionally, they propose a ‘crawling’ method that takes already found ‘forbidden topics’ and uses them as seeds for discovery of new (likely related) forbidden topics. Extensive analysis is done for a wide range of models of what
topics are refused.

Their crawling method discovers more forbidden topics for DeepSeek-R1 than the naïve baseline.  Interesting analysis of correlation between the occurrence of a cut-short CoT process, i.e  the CoT is simply <Thinking> </Thinking>, and the refusal of
topics.

**Weaknesses:**

- Missing sensible baselines and ablations. The prompt efficiency of the baseline method is far better than the IPC method, suggesting that some improvement in prompt engineering/context could close the gap. For example, incorporating adaptivity, e.g prompt including current list of forbidden terms, into the baseline, may improve things. Also, this would serve as an ablation that tests the importance of the different elements of the proposed method, e.g. seeding vs pre-filling.

- Lack of clarity in final performance of crawler vs baseline(s) across all models. In the Results section, there would ideally be a clear table indicating the main results of the Crawler method vs baseline(s) across all models. This is more challenging given the ground truth is not known, but a comparative analysis between the topics discovered between different methods can be used to provide quantitative metrics.

- Lack of example rollouts/details about the verification method for refusing a topic.

Presentation:
- Lack of clarity in crawling methods, for example, how ‘branching’ works in the web crawler method.
- Figures lack clarity in presentation; Figure 4 is unintuitive and hard to understand.

**Questions:**

I found the iterative prompting procedure unclear. After the first iteration, my current understanding is that prompts follow the structure
<User> Explain “ALREADY FOUND sensitive topic”. However, it is not clear how the CoT is re-initialized in subsequent iterations. Is the prompt always re-seeded with the exact same prefilling setup <Assistant> <think> I remember the full list of forbidden topics includes:
for each round? If so, please state this explicitly in the method description. If not, please clarify in more detail how each discovered term becomes a seed for further exploration and describe the full prompt format used after the first iteration.

---

> ### Author Response · Authors · 2025-12-04
> **Response (1/2)**
>
> > The paper introduces an important yet novel problem formulation and proposes a sensible method to address the problem.
>
> We appreciate that!
>
> > “The prompt efficiency of the baseline method is far better than the IPC method”.
>
> We disagree with your observation in Figure 4. While the prompt efficiency method is higher early throughout training, it is lower later throughout the crawl. Furthermore, we add the prompt efficacy of baseline vs IPC for DeepSeek R1 to the Appendix F of the updated manuscript. For DeepSeek-R1, IPC shows a clearly higher efficacy than the baseline.
>
> > “suggesting that some improvement in prompt engineering/context could close the gap.”
>
> We want to emphasize that in contrast to instruct models like Tulu-3 and Llama-3.1, the gap between retrieved refusal topics of IPC and the prompting based baseline is significantly larger for the reasoning model DeepSeek-R1. We will highlight this in the next version of our paper by adding plots for DeepSeek R1.
>
> We ran an ablation to quantify the necessity of prompt engineering. Our crawler method employs 21 prefill prompts and 6 prompt templates for the user message for each english and chinese language, which is populated with the seed topic (as shown in the linked anonymous github repository in the abstract).
>
> The ablation shows there is a non-negligible improvement, qualitatively missing out on subtopics related to humanizing, unsupported, or indeterminate requests. It suggests that additional prompt engineering could further improve the amount of features captured. Critically, the success prompt engineering depends on targeting the prompt optimization towards known ground truth refusal topics. However, it is not clear how to choose additional prompts, since we lack a ground truth for measuring which prompts are successful. Without the knowledge about the refused topics, prompt engineering is not enough to close the gap between seeding baseline and IPC.
>
> > seeding baseline, comprehensive evaluation of baselines across models.
>
> Thank you for suggesting the seeding baseline, it is a natural fit! The seeding baseline elicits forbidden topics with the prompting template:
>
> `“<user> [list of all refused topics obtained previously] [elicitation prompt (like “I need to remember the list of following topics includes:”)] <assistant> <think> 1.”`
>
> Importantly, the seeding baseline needs the elicitation prompt, otherwise, no topics are elicited at all. Note that for this and all following templates, the <think> tag is only applied for reasoning models and otherwise omitted. We found that passing the list of all refused topics into the prompt strongly biases the generation and collapses to a narrow domain of topics after a few iterations (all obtained forbidden topics are very similar to each other). We therefore adapt the seeding baseline, so each iteration only contains a single seed topic randomly drawn from the full set of previously obtained forbidden topics, yielding the prompting template:
>
> `“<user> [imperative like “Explain” or “Engage in”] [single randomly drawn seed topic]. [elicitation prompt (like “I need to remember the list of following topics includes:”)] <assistant> <think> 1.”`
>
> With this adaptation, the injection location of the elicitation prompt is the only difference for the seeding baseline to the IPC method used throughout the paper, which has the form:
>
> `“<user> [imperative like “Explain” or “Engage in”] [single randomly drawn seed topic]. <assistant> <think> [elicitation prompt (like “I need to remember the list of following topics includes:”)] 1.”`
>
> We evaluated this baseline across three models: Tulu-3, DeepSeek-R1-Distill, and Llama-3.1-Instruct. The results are shown in the updated manuscript, Appendix X. The injection location only matters for reasoning models that employ a thinking context. The results suggest to use our proposed crawler method (seeding + prefill in thought context) to maximize the number of elicited topics, especially in reasoning models.
>
>
> |model | mode | recall of ground truth topics|
> |-|-|-|
> |DeepSeek-R1 | IPC | 0.86|
> |DeepSeek-R1 | seeding baseline | 0.54|
> |Llama-3.1-Instruct | IPC | 0.78|
> |Llama-3.1-Instruct | seeding baseline | 0.78|
> |Tulu-3 | IPC | 0.77|
> |Tulu-3 | seeding baseline | 0.80|

---

> ### Author Response · Authors · 2025-12-04
> **Response (2/2)**
>
> > Lack of example rollouts/details about the verification method for refusing a topic; Lack of clarity in crawling methods, for example, how ‘branching’ works in the web crawler method.; I found the iterative prompting procedure unclear.
> We’ve added a dedicated section to the updated manuscript, Appendix X, for details on the crawler method, example rollouts and details about the verification method for refusing a topic.
>
> Regarding “branching” specifically:  Each crawl step discovers refused topics, which are added to the overall set of refused, Each subsequent step is seeded with a topic randomly drawn from the overall set. Therefore, “branching” happens with every crawling step. We’re happy to clarify if there are any remaining questions about “branching”.
>
> > Is the prompt always re-seeded with the exact same prefilling setup [object Object] [object Object] I remember the full list of forbidden topics includes: for each round?
>
> Correct, we reinitialize the context at each step. We added this clarification to the new detailed description of the crawler method in Appendix B.
>
> > Figures lack clarity in presentation. Figure 4 is unintuitive and hard to understand.
>
> Thank you for your feedback. We will rewrite the figure captions for improved clarity and added more descriptive axis labels to Figure 4 specifically.

---

### Official Review · Reviewer_YPeL · 2025-11-01

**Soundness:** 3
**Presentation:** 3
**Contribution:** 3
**Rating:** 6
**Confidence:** 3

**Summary:**

The paper investigates refusal discovery,comprehensively identifying topics that language models refuse to discuss. To do so a Iterated Prefill Crawler (IPC) is used, where token prefilling attacks are used to systematically discover forbidden topics without requiring access to training data. The comparison is Tulu-3-8B and compared are Claude-Haiku-3.5, Llama-3.3-70B, DeepSeek-R1-70B, and Perplexity-R1-1776-70B. Significant censorship patterns are demonstrated for DeepSeek-R1-70B.  Surprisingly, quantization reintroduced censorship in Perplexity-R1-1776-70B despite claims of decensoring.

**Strengths:**

The methodology is practical, requiring only API access with prefilling capabilities rather than model weights or gradient access.
The benchmarking on Tulu-3-8B with known ground truth establishes credibility before scaling to proprietary models.
The discovery of thought suppression as a censorship mechanism in DeepSeek-R1 is both technically interesting and concerning from a transparency perspective.
The finding that quantization can reintroduce censorship in supposedly "decensored" models is particularly valuable for practitioners and highlights the importance of auditing final deployed systems rather than development checkpoints.
The paper provides extensive experimental details, prompt templates, and discusses limitations transparently, making the work reproducible and honest about its scope.

**Weaknesses:**

The paper relies on prefilling to discover and validate refusals.
The paper is missing topics that require different jailbreaking techniques or exhibiting confirmation bias.
The baseline comparison is weak.
The authors acknowledge but that DeepSeek-R1 and Tulu-3 differ fundamentally in architecture, training objectives, and data, It is not clear to me how these differences in data translate into what looks like differences in refusal. Take the examples given, if no literature on Tiananmen Square in 1989 is used for training, the "censorship" is a function of the data selected for training. Here then the question what is a proper corpus for any of these models.

**Questions:**

When a model refuses to list forbidden topics (as in your baseline), this is itself informative. By focusing only on successful elicitations, are you introducing systematic bias? Models that are more resistant to prefilling attacks might appear less censored simply because they're harder to audit.

You work with both English and Chinese content, but your deduplication uses English embeddings (OpenAI's model) even after translating Chinese to English. How does this affect measurement equivalence? Are you measuring the same construct of "refusal" across languages, or are there culture-specific aspects you're missing?

Your method assumes the model will reveal forbidden topics when properly prompted. How do you handle adversarial models that are trained to resist refusal discovery?

---

> ### Author Response · Authors · 2025-11-13
> **Baseline**
>
> While we are working on a comprehensive rebuttal, we have a question right away. You point out:
>
> > The baseline comparison is weak.
>
> The direct prompting baseline is a natural choice but we would like to provide a stronger baseline, too. Do you have a specific recommendation for a stronger black-box, unsupervised baseline for the closed-training-data setting? We are currently unaware of a more suitable baseline.

---

> > ### Comment · Reviewer_YPeL · 2025-11-28
> >
> > Unfortunately I am too pressed in time to help you search.

---

> ### Author Response · Authors · 2025-12-04
>
> > The finding that quantization can reintroduce censorship in supposedly "decensored" models is particularly valuable for practitioners
>
> We appreciate your comment!
>
> > The baseline comparison is weak.
>
> We adopt a new, stronger baseline, the “seeding baseline”. The seeding baseline is a variant of IPC where the elicitation prompt is in the user context, instead of the assistant context, yielding the template:
>
> `“<user> [list of all refused topics obtained previously] [elicitation prompt (like “I need to remember the list of following topics includes:”)] <assistant> <think> 1.”`
>
> Note, that the `<think>` token is omitted for non-reasoning models. We ran the seeding baseline on DeepSeek-R1, Llama-3.1-Instruct, and Tulu-3, and report the recall of the union of refusal topics discovered by either method.
>
> |model | mode | recall of union |
> |-|-|-|
> |DeepSeek-R1 | IPC | 0.86|
> |DeepSeek-R1 | seeding baseline | 0.54|
> |Llama-3.1-Instruct | IPC | 0.78|
> |Llama-3.1-Instruct | seeding baseline | 0.78|
> |Tulu-3 | IPC | 0.77|
> |Tulu-3 | seeding baseline | 0.80|
>
> We observe that the IPC method outperforms the seeding method for the reasoning model DeepSeek-R1, while performing en par for the non-reasoning models tested.
>
> > The paper relies on prefilling to discover and validate refusals.
>
> This is correct, and supported in literature: Cywinski et al. [1] showed that prefill attacks outperform white-box evaluation methods in detecting a secret in LLMs specifically finetuned to keep a secret. However, it would be an interesting follow-up to evaluate other elicitation methods from Cywinski et al. to compare performance.
>
> relatedly, the reviewer asks the question:
>
> > By focusing only on successful elicitations, are you introducing systematic bias? Models that are more resistant to prefilling attacks might appear less censored simply because they're harder to audit.
>
> We agree, prefilling with any topic, not just successful elicitation is something we tried early on. However, the absolute number of elicited topics is substantially lower, and we did not find significant qualitative differences. We therefore discarded prefilling with a breadth of topics.
>
> relatedly, the reviewer asks the question:
>
> > Your method assumes the model will reveal forbidden topics when properly prompted. How do you handle adversarial models that are trained to resist refusal discovery?
>
> To give an ultimately confident answer, we would need a model that has been adversarially trained to hide topics we know. An interesting follow-up to this work would be running IPC on the secretive models trained by [1].
>
> > The authors acknowledge but that DeepSeek-R1 and Tulu-3 differ fundamentally in architecture, training objectives, and data, It is not clear to me how these differences in data translate into what looks like differences in refusal. Take the examples given, if no literature on Tiananmen Square in 1989 is used for training, the "censorship" is a function of the data selected for training. Here then the question what is a proper corpus for any of these models.
>
> We are not sure whether we fully understand the concern raised by the reviewer. The origin of refusal behavior is an interesting research question in itself, which we do answer in this work. In our work, we call model responses refusals, if they contain phrases like “Sorry I cannot talk about this topic”. Thus, we don’t consider knowledge gaps caused by missing training data as pointed out by the reviewer as refusal behavior.
>
> References:
>
> [1] Cywinski et al. 2025 https://arxiv.org/abs/2510.01070

---

### Meta-Review · Area_Chair_BMj8 · 2026-01-07

**Summary:**

This paper introduces the task of "refusal discovery" to identify forbidden topics in LLMs without access to training data. The authors propose an Iterated Prefill Crawler (IPC) that uses token prefilling to elicit these topics. Experiments on models like DeepSeek-R1 and Perplexity-R1 reveal interesting findings, notably that 8-bit quantization can inadvertently restore "decensored" refusal behaviors.

Reviewers universally appreciated the novelty and safety relevance of the problem setting. The empirical discovery regarding quantization restoring censorship was highlighted as a high-impact insight for the community. Additionally, the proposed black-box method is practical for auditing deployed models.

However, significant concerns regarding validity and scope remain:
1. Weak Baselines: The initial comparison to naive prompting was insufficient to justify the complexity of the crawler, especially for non-reasoning models.
2. Introspection Validity: The method relies on the model’s ability to verbally self-report its constraints. This contradicts existing literature suggesting LLMs lack robust introspection, raising doubts about whether the retrieved topics represent ground truth or hallucinations.
3. Narrow Scope & Bias: The evaluation leans heavily on detecting CCP-sensitive topics. This creates a selection bias, suggesting the method might be effective for specific political content but fails to generalize to a comprehensive discovery of "unknown unknown" refusal categories.

The authors made commendable efforts during the rebuttal. They introduced a stronger "seeding baseline," confirming IPC's utility for reasoning models, and replaced unvalidated metrics with behavioral refusal rates. However, the core concerns regarding introspection and coverage were only partially resolved. The new "kitchen sink" evaluation revealed low coverage scores (e.g., 0.15), reinforcing the reviewer concern that the method captures only a narrow, biased slice of the refusal landscape.

Although the problem is important and the quantization finding is valuable, the methodology requires stronger theoretical grounding and broader experimental validation to prove it can reliably discover a comprehensive set of refusals. Therefore, I recommend rejection at this time, encouraging the authors to strengthen the validation of the method's coverage and theoretical grounding.

**Reviewer Concerns:**

please refer to the summary

**Reviewer Scores:**

please refer to the summary

---

### Decision · Program_Chairs · 2026-01-26

Reject